# Lipid Microenvironment Modulates the Pore-Forming Ability of Polymyxin B

**DOI:** 10.3390/antibiotics11101445

**Published:** 2022-10-20

**Authors:** Anastasiia A. Zakharova, Svetlana S. Efimova, Olga S. Ostroumova

**Affiliations:** Institute of Cytology of Russian Academy of Sciences, Tikhoretsky 4, 194064 Saint Petersburg, Russia

**Keywords:** polymyxin B, ion channel, lipid bilayers, lipopolysaccharides, Kdo_2_-Lipid A, toroidal pore, membrane dipole potential, phloretin

## Abstract

The ability of polymyxin B, an antibiotic used to treat infections caused by multidrug-resistant Gram-negative bacteria as a last-line therapeutic option, to form ion pores in model membranes composed of various phospholipids and lipopolysaccharides was studied. Our data demonstrate that polymyxin B predominantly interacts with negatively charged lipids. Susceptibility decreases as follows: Kdo_2_-Lipid A >> DOPG ≈ DOPS >> DPhPG ≈ TOCL ≈ Lipid A. The dimer and hexamer of polymyxin B are involved in the pore formation in DOPG(DOPS)- and Kdo_2_-Lipid A-enriched bilayers, respectively. The pore-forming ability of polymyxin B significantly depends on the shape of membrane lipids, which indicates that the antibiotic produces toroidal lipopeptide-lipid pores. Small amphiphilic molecules diminishing the membrane dipole potential and inducing positive curvature stress were shown to be agonists of pore formation by polymyxin B and might be used to develop innovative lipopeptide-based formulations.

## 1. Introduction

The emergence of multidrug-resistant Gram-negative bacteria prompted clinicians to change their minds regarding the assessment of the toxicity of polymyxin B (PMB), an *N*-acylated cyclic decapeptide from *Paenibacillus polymyxa*, and reintroduce it into clinical practice as a “last-line” therapy [1,2]. Despite the lack of activity against Gram-positive bacteria and anaerobes, PMB can still be used against a variety of Gram-negative bacilli, including most clinically relevant species such as *Escherichia coli*, *Enterobacter* spp., *Klebsiella* spp., *Citrobacter* spp., *Salmonella* spp., and *Shigella* spp. [3,4,5,6,7,8,9]. In addition, PMB is effective against non-fermentative Gram-negative pathogens such as *Acinetobacter baumannii* and *Pseudomonas aeruginosa*. Although the antimicrobial activity of the lipopeptide has been studied for more than 50 years, the detailed mechanisms of action of the antibiotic are not fully understood. The PMB chemical structure is presented on Figure 1.

Early studies showed that Gram-negative bacteria pretreated with PMB became more susceptible to lysozyme treatment, indicating that it disrupted the outer membrane, forming electron microscopy-visible protrusions and thereby exposing the underlying peptidoglycan layer to lysozymes [10,11]. Additionally, the treatment of *E. coli* with PMB increased susceptibility to beta-lactam antibiotics, which target the peptidoglycan synthesis machinery, in the bacterial species [12]. Lipopolysaccharides are predominant lipid species in the outer membrane of Gram-negative bacteria. The lipid base of the lipopolysaccharides is represented by lipid A (a phosphorylated glucosamine dimer with four to seven saturated acyl chains), most often linked to a core oligosaccharide through two residues of 3-deoxy-d-manno-octulosonic acid (Kdo) (the structures of the two lipopolysaccharides used in this study are presented on Figure 1). A model of PMB binding to lipopolysaccharides suggesting the formation of complex stabilized by a combination of electrostatic and hydrophobic interactions has been proposed [13,14,15,16,17,18,19,20,21,22,23]. It was shown that the cationic α,γ-diaminobutyric acid residues in the polypeptide part of PMB molecules bind to the negatively charged Kdo residues of lipopolysaccharides while the PMB fatty acid group interacts to the hydrocarbon tails of lipid A [19,22]. Molecular dynamic simulations carried out by Santos et al. [24] revealed that the effects of PMB on bilayer thickness and curvature depend on the chemotype of the lipopolysaccharide membrane.

It is believed that PMB can be transported via the outer membrane by a self-promoted uptake mechanism [22,23,25,26,27,28,29]. Using asymmetric lipid bilayers, Shroder et al. [30] showed that the incorporation of PMB into a lipopolysaccharide monolayer led to an appearance of the discrete fluctuations of the membrane conductance. The authors concluded that the aggregation of a large number of lipopeptide molecules was accompanied by a local disordering and the formation of transmembrane pore-like lesions. Moreover, after comparing the diameter of the cross-sectional area of the PMB peptide ring (about 1.23 nm^2^) and the lipopeptide-induced lesions (approximately 3 nm^2^), Shroder et al. [30] theorized a diffusion of the antibiotic across the self-made transmembrane pores.

Whether the permeabilization of the outer membrane is responsible for the antimicrobial activity of PMB is still under debate [31]. Perhaps the penetration of PMB through the outer membrane is just the first step in reaching other targets, in particular, the inner bacterial membrane that is composed of phospholipids. The significant role of phospholipids in lipopeptide antimicrobial action is confirmed by the lipidome analysis of PMB-resistant Gram-negative bacteria. PMB-resistant *Pseudomonas aeruginosa* have shown increasing contents of cardiolipin and diminishing contents of phosphatidylglycerol and phosphatidylethanolamine [32]. Tao et al. [33] linked the decrease in phosphatidylethanolamine percentage with lipopolysaccharide deficiency. Imai et al. [34] demonstrated that acidic phospholipids are necessary to confer the susceptibility of artificial liposomes to PMB. These facts are consistent with an assumption of lipopeptide–phospholipid interaction. The antibiotic-induced changes in the properties of phospholipid membranes are also in agreement with this assumption. In particular, PMB adsorption onto the surface of phosphatidylserine monolayers reduces their capacity and increases ion conductance [35]. The incorporation of PMB in negatively charged vesicles composed of phosphatidylcholine and phosphatidylglycerol leads to membrane disordering [36]. On the contrary, PMB stiffens bilayers made from a mixture of phosphatidylglycerol with phosphatidylethanolamine, thus increasing the lipid tail order and membrane bending rigidity [37]. PMB was also shown to induce lipid segregation in lecithin/phosphatidic acid mixtures [38]. Table 1 summarizes the membrane-associated effects of PMB that determine its antibacterial activity. More complete reviews of the modes of PMB antimicrobial action can be found in [39,40,41,42].

Thus, a detailed study of the lipid specificity of pore formation by PMB and a search for a way to modulate it are important steps in the development of more effective drug formulations. Here, we studied the features of the pore formation of PMB in planar lipid bilayers composed of different phospholipids and lipopolysaccharides. An analysis of the concentration dependences of the PMB-induced transmembrane macroscopic current revealed a number of lipopeptide molecules involved in the formation of the conductive subunits in membranes of various compositions. Regulatory pathways and small molecules potentiating the pore-forming ability of PMB were also revealed.

## 2. Results and Discussion

### 2.1. Pore-Forming Activity of Polymyxin B in Phospholipid Bilayers

It is believed that cationic PMB targets the membranes of pathogenic microorganisms primarily due to electrostatic interactions with negatively charged lipids, in particular, lipopolysaccharides in the outer membrane and phosphatidylglycerol and cardiolipin in the inner bacterial membrane. Confirming this, the one-side addition of PMB into a solution of bathing bilayers composed of uncharged dioleoylphosphocholine (DOPC, Figure 1) up to 60 µM did not lead to an increase in ion permeability (Figure 2a). Further enlargement in the lipopeptide concentration in the membrane bathing solution was accompanied by a loss of the electrical stability of the DOPC bilayers and their subsequent destruction. These data were in agreement with data of Domingues et al. [49] that suggested a destabilization of palmitoyoleoylphosphocholine membranes by the electrostatic repulsion between the positively charged terminal amino group of the choline and the Dab residues of PMB.

The replacement of DOPC with negatively charged dioleoylphosphoglycerol (DOPG, Figure 1) led to the appearance of discrete step-like current fluctuations of various amplitudes related to the openings and closures of single PMB pores (Figure 2b, inset). The threshold lipopeptide concentration required to observe single-channel activity was 10 μM. This value exceeds the minimum inhibitory concentration of PMB against *E. coli* and *A. baumannii* [50] by 25 times. This fact does not allow us to associate the antibacterial action of the lipopeptide with pore formation in phosphatidylglycerol-enriched inner bacterial membranes. Figure 2b presents an example of the time track of the PMB-induced current flowing through a DOPG membrane resulting from the subsequent lipopeptide additions. An analysis of the dependence of the steady-state transmembrane current induced by PMB on its concentration revealed that lipopeptide did not produce any changes in bilayer ion permeability at the concentration range of 2.5 ÷ 5 µM (Figure 3a, squares). An increase in PMB concentration from 5 to 100 µM caused an enlargement of the macroscopic transmembrane current. The dose-dependence of the current tended to saturate at antibiotic concentrations above 100 µM. A linear regression of the growth region of the *I*(*C*)-bilogarithmic plot gives a slope (*m*) close to 2 (Figure 3a, squares). This indicates that the transmembrane current increased with an approximately second power of the lipopeptide concentration. Sheppard et al. [51] reported that the macroscopic conductance induced by the cyclic lipopeptide from *Bacillus subtilis*, surfactin was also enhanced with the second power of lipopeptide concentration. The authors concluded that surfactin dimers were involved in the formation of functional channel [51]. A similar conclusion was reached by studying the dose-dependence of the transmembrane current induced by another cyclic lipopeptide from *B. subtilis*, fengycin [52].

A decrease in the membrane content of DOPG up to 50 mol% did not lead to significant changes in the pore-forming activity of PMB compared with the DOPG bilayers (Figure 2c). Initial single pore events occurred at 5 µM of lipopeptide (Figure 2c and Figure 3a, circles). Similar to the DOPG membranes, an increase in the PMB concentration from 5 to 90 µM in the solution bathing the DOPC/DOPG bilayers was accompanied by an increase in the lipopeptide-induced transmembrane current. A further increase in the antibiotic dose (more than 90 µM) did not lead to an increase in the transmembrane current. Fitting the growth region of the *I*(*C*)-dependence in bilogarithmic coordinates with the linear function provided the number of PMB molecules leading to pore formation in the DOPC/DOPG membrane: 2. Considering that the incorporation of DOPC into the DOPG membrane had practically no effect on the pore-forming ability of PMB, the obtained data did not contradict the assumption that the cationic lipopeptide predominantly interacted with negatively charged phosphatidylglycerol.

The replacement of DOPG with dioleoylphosphoserine (DOPS, Figure 1) did not significantly alter the pore-forming activity of the lipopeptide (Figure 2d and Figure 3a, diamonds), indicating that the type of the negatively charged lipid head group was not of key importance. The PMB concentrations required to form pores in the DOPS bilayers (5–100 µM) were less than the antibiotic IC_50_ for different cell lines (300–1000 µM [53,54]). Thus, the ability of the lipopeptide to bind to DOPS and form ion-conductive pores in the DOPS-enriched membranes might be responsible for its high toxicity.

The replacement of DOPG with cone-shaped lipids, diphytanoylphosphoglycerol (DPhPG, Figure 1) (having branched acyl chains) or cardiolipin (TOCL, Figure 1) (having four tails), significantly inhibited the ability of PMB to produce transmembrane pores (Figure 2e,f, respectively). The revealed dependence of the pore-forming ability of the antibiotic on the shape of lipid molecules might indicate that PMB forms toroidal lipopeptide-lipid pores. The formation of a water/ion-permeable hole in a membrane requires the lining of the transport pathway by hydrophilic regions of the molecules involved in its formation, polar or charged amino acid residues, and hydrophilic lipid heads. This model assumes the incorporation of lipopeptide molecules into the hydrophilic region of the adjacent lipid monolayer and the bending of opposite lipid monolayer to fully perforate the bilayer. It has been successfully used to describe the architecture of the pores formed by the cyclic lipopeptide from *Pseudomonas syringae*, syringomycin E [55]. It is believed that the pores of this type have a toroidal geometry and lipid mouth of positive curvature, and the compounds that increase the spontaneous curvature should contribute to their formation while those that increase the negative curvature stress should inhibit the formation of the pores due to the high cost of forming a lipid mouth [56,57].

### 2.2. Pore-Forming Activity of Polymyxin B in Lipopolysaccharide-Enriched Model Membranes

Considering that the lipopolysaccharides are the major targets of PMB action, we analyzed the ability of PMB to induce ion-permeable pores in the DOPC/DOPG membrane containing 1 mol% of di[3-deoxy-D-manno-octulosonyl]-lipid A (Kdo_2_-Lipid A, Figure 1). Figure 4a presents a sample record of current flowing through the PMB-modified DOPC/DOPG/Kdo_2_-Lipid A (49.5/49.5/1 mol%) bilayer. The addition of lipopeptide of up to 1 μM into the bilayer-bathing solution led to the appearance of a large number of step-like transmembrane current fluctuations of different amplitudes (Figure 4a, insert). The subsequent increase in PMB concentration from 1.5 to 5 μM resulted in a gradual enlargement of the macroscopic transmembrane current and tended to already saturate at 10 μM (Figure 4a). Fitting the growth region of the dependence of the PMB-induced transmembrane current on lipopeptide concentration in bilogarithmic coordinates with the linear function indicated a slope factor (*m*) of about 6 (Figure 3b, triangles). This indicated the significant increase in the cooperativity of PMB binding to the lipopolysaccharide-enriched bilayers compared with phospholipid membranes and the involvement of 5 ÷ 9 antibiotic molecules in the formation of PMB pores in the DOPC/DOPG/Kdo_2_-Lipid A (49.5/49.5/1 mol%) membranes. It should also be noted that the macroscopic current induced by PMB in the DOPC/DOPG/Kdo_2_-Lipid A (49.5/49.5/1 mol%) membranes was two orders higher than that in the DOPC/DOPG (50/50 mol%) bilayers (Figure 3a vs. Figure 3b). Considering that the amplitude of single PMB pores varied in the picoampere range regardless of the lipopolysaccharide content (Figure 2c and Figure 4a), these results indicated a dramatic difference in the number of lipopeptide pores in the membranes without and with Kdo_2_-Lipid A. These data are in agreement with the results of a molecular dynamics simulation study performed by Berglund et al. [58]. This was also evidenced by the 10-fold difference in the threshold PMB concentration (Figure 3a,b).

The exclusion of DOPG from the Kdo_2_-Lipid A-enriched bilayers was accompanied by a 2-fold enlargement in the PMB threshold concentration, although this had no effect on the macroscopic lipopeptide-induced current and the slope of the straight line fitting the growth region of *I*(*C*)-bilogarithmic plot (Figure 3b, circles). This means that PMB preferentially interacts with Kdo_2_-Lipid A, but its adsorption on the membrane (and, consequently, PMB membrane concentration) depends on the bilayer content of the negatively charged phospholipid. Confirming this issue, the probability of the formation of PMB pores in the DOPC/Kdo_2_-Lipid A (99/1 mol%) membranes was decreased compared with the DOPC/DOPG/Kdo_2_-Lipid A (49.5/49.5/1 mol%) bilayers.

The incorporation of cone-shaped phospholipids into the lipopolysaccharide-enriched membranes (the replacement of DOPC and DOPG with diphytanoylphosphocholine (DPhPC) and DPhPG, respectively) did not affect the stoichiometry of the PMB binding to the bilayer, but it did lead to a 10-fold reduction in the macroscopic PMB-induced current and an approximate 4-fold increase in threshold lipopeptide concentration (Figure 4b and Figure 3b, squares). These data were in agreement with the assumption that a negative spontaneous curvature inhibited the formation of toroidal PMB-lipid pores in both the phospholipid and Kdo_2_-Lipid A-enriched bilayers.

It is widely recognized that the remodeling of lipopolysaccharides is a major survival strategy for Gram-negative bacteria [59,60,61]. Therefore, we analyzed the dependence of PMB’s ability to form ion-permeable pores on the lipopolysaccharide chemotype. Figure 4c shows that replacement of Kdo_2_-Lipid A with its penta-acylated analogue without sugar residues, Lipid A (Figure 1), significantly inhibited the ability of PMB to produce transmembrane pores. The lipopeptide was not able to induce macroscopic current flowing through the DOPC/DOPG/Lipid A (49.5/49.5/1 mol%) bilayers, as only single PMB pores were observed at lipopeptide concentrations over 40 µM. The destruction of the DOPC/DOPG/Lipid A (49.5/49.5/1 mol%) membranes in the presence of PMB occurred at concentrations of more than 100 μM. These results were in agreement with data of a molecular dynamic study performed by Santos et al. [24] that demonstrated the key roles of both the Kdo-modification and hexa-acylation of Lipid A in susceptibility to PMB.

### 2.3. Alteration in Polymyxin B Pore-Forming Activity in the Presence of Small Molecules Modulating Membrane Physical Properties

The observed dependence of the PMB pore-forming activity on the charge and shape of membrane lipids indicated that the formation of pores by lipopeptide is altered by the transmembrane distribution of the electrical potential (especially membrane surface charge and potential) and the lateral pressure (which depends on the shape of bilayer-forming molecules). This means that changing these properties with non-toxic small molecules might be a way to up-regulate the ability of the antibiotic to form pores, i.e., to reduce its therapeutic concentration and toxicity. Firstly, we estimated the effects of alteration in the membrane boundary potential by small molecules on the pore-forming ability of PMB. An addition of local anesthetic tetracaine into the membrane bathing solution up to 0.5 mM led to an approximate 3-fold decrease in the steady-state PMB-induced current (Figure 5a, tetracaine). This effect might be referred to as an increase in the surface potential by positively charged anesthetic molecules [62,63]. The adsorption of positively charged tetracaine molecules on the membrane might counteract the adsorption of positively charged lipopeptides (Figure 6, step I, arrow to the left). As shown above (Figure 2b–d), the introduction of negatively charged lipids into the membrane composition produced the opposite effect (Figure 6, step I, arrow to the right). Interestingly, in addition to the alteration in surface potential, tetracaine is known to increase the membrane dipole potential, i.e., the positive potential in the hydrophobic core of the lipid bilayer [63,64]. The growth of dipole potential should be accompanied by an increase in the energy of the immersion of positively charged PMB molecules into a membrane and a decrease in its pore-forming ability (Figure 6, step II, arrow to the left). Supporting this, an introduction of 10 µM of styryl dye, RH 421, known to increase membrane dipole potential [65,66], led to an approximate 2-fold decrease in the macroscopic current flowing through the PMB-modified DOPC/DOPG/Kdo_2_-Lipid A (49.5/49.5/1 mol%) membrane (Figure 5a, RH 421). To reduce the effective concentration and, consequently, toxicity of the antibiotic, the compounds that are able to diminish the dipole potential of the membrane and thereby able to potentiate the pore-forming activity of PMB are of the greatest interest. Figure 5 illustrates that plant polyphenol (phloretin) and an inhibitor of phosphodiesterase type 5 (vardenafil) known to reduce membrane dipole potential [67,68,69] significantly enhanced the PMB-induced transmembrane current (by about 30 and 40 times at 20 and 100 µM, respectively). Table 2 clearly demonstrates the ratio between the lipopeptide-induced steady-state transmembrane current before and after the addition of the small molecule (*I_∞_*/*I_∞_^0^*) to compound-induced changes in the boundary potential of the DOPC/DOPG/Kdo_2_-Lipid A (49.5/49.5/1 mol%) membranes (∆*ϕ_b_*): the boundary potential decreased, the current increased, and vice versa.

To confirm that phloretin potentiates the action of PMB on the lipid bilayers mimicking the outer membrane of Gram-negative bacteria, we performed a confocal fluorescence microscopy study. Figure 7 presents the typical fluorescence micrographs of liposomes composed of palmitoyloleoylphosphocholine/palmitoyloleoyl-phosphoglycerol/Kdo_2_-Lipid A (79/20/1 mol%) in the absence (Figure 7a) and presence of 5 µM of PMB (Figure 7b). The addition of the antibiotic led to the aggregation of vesicles and a significant decrease in their size. The combination of 5 μM of PMB and 400 μM of phloretin caused the complete destruction and aggregation of collapsed lipid vesicles, demonstrating that phloretin enhanced the membrane lytic properties of PMB (Figure 7c).

Variation in membrane lipid composition indicated the possibility of the formation of toroidal lipopeptide-lipid pores by PMB. According to the model, the incorporation of lipids or molecules with an inverted cone shape, inducing positive spontaneous curvature, should potentiate the formation of toroidal lipopeptide/peptide-lipid pores (Figure 6, step II, arrow to the right), while the incorporation of lipids or molecules, having a conical shape and inducing negative spontaneous curvature, should inhibit the formation of toroidal lipopeptide/peptide-lipid pores (Figure 6, step II, arrow to the left) [70,71,72,73,74,75]. Figure 3b demonstrates that palmitoylphosphocholine (LysoPC), having only one acyl chain and producing positive curvature stress [76,77], enhanced the PMB pore-forming activity by about 40 times at 7.5 µM, while the addition of oleic acid of up to 30 µM, known to produce negative curvature stress [70] similar to DPhPC and TOCL, led to a 5-fold decrease in the steady-state transmembrane current induced by PMB (Table 3). Table 3 shows that the increase in the spontaneous radii of curvature (*R*) led to an increase in the lipopeptide-induced steady-state transmembrane current (*I_∞_*). Thus, the obtained data were in good agreement with the results regarding lipid composition variation and the assumption that PMB forms toroidal lipopeptide-lipid pores.

## 3. Materials and Methods

### 3.1. Chemical Reagents

Nonactin, phloretin, vardenafil, RH 421, tetracaine hydrochloride, methyl-β-cyclodextrin, KCl, HEPES, pentane, ethanol, DMSO, sorbitol, and polymyxin B (PMB) were purchased from Sigma-Aldrich Company Ltd. (Gillingham, United Kingdom). Solutions of 0.1 M KCl were buffered using HEPES-KOH at pH 7.4. The 1,2-dioleoyl-*sn*-glycero-3-phosphocholine (DOPC), 1-palmitoyl-2-oleoyl-glycero-3-phosphocholine, 1,2-dioleoyl-*sn*-glycero-3-phospho-(1’-rac-glycerol) (sodium salt) (DOPG), 1-palmitoyl-2-oleoyl-*sn*-glycero-3-phospho-(1’-rac-glycerol), 1,2-dioleoyl-*sn*-glycero-3-phospho-L-serine (sodium salt) (DOPS), 1’,3’-bis[1,2-dioleoyl-*sn*-glycero-3-phospho]-glycerol (sodium salt) (TOCL), 1,2-diphytanoyl-*sn*-glycero-3-phospho-(1’-rac-glycerol) (sodium salt) (DPhPG), 1,2-diphytanoyl-*sn*-glycero-3-phosphocholine (DPhPC), Di[3-deoxy-D-manno-octulosonyl]-lipid A (ammonium salt) (Kdo_2_-Lipid A), detoxified lipid A from *Salmonella minnesota* R595 (Lipid A), 1-palmitoyl-2-hydroxy-*sn*-glycero-3-phosphocholine (LysoPC), oleic acid, and 1,2-dipalmitoyl-sn-glycero-3-phosphoethanolamine-*N*-lissamine rhodamine (Rh-DPPE) lipids were obtained from Avanti Polar Lipids^®^. The chemical structures of the lipids used in the study are presented in Figure 1.

All experiments were performed at room temperature (25 °C).

### 3.2. Studying PMB Pore-Forming Ability in Planar Lipid Bilayers

Virtually solvent-free planar lipid bilayers were formed with a monolayer-opposition technique [79] at an aperture of 50 µm in diameter, with a 10 µm thick Teflon film separating the two (*cis*- and *trans*-) compartments of the Teflon chamber. The aperture was pretreated with hexadecane. Lipid bilayers were prepared from pure DOPC, DOPS, DOPG, TOCL, or DPhPG and mixtures of DOPC/DOPG (50/50 mol%), DOPC/DOPG/Kdo_2_-Lipid A (49.5/49.5/1 mol%), DPhPC/DPhPG/Kdo_2_-Lipid A (49.5/49.5/1 mol%), DOPC/Kdo_2_-Lipid A (99/1 mol%), and DOPC/DOPG/Lipid A (49.5/49.5/1 mol%). Membranes were bathed in 0.1 M KCl and 10 mM HEPES at pH 7.4. After the lipid bilayer was completely formed and stabilized, PMB from the 7.2 mM aqueous stock solution was added to *cis*-side of the Teflon chamber to obtain a final concentration in the range of 0.5 to 100 μM.

To apply the transmembrane voltage (*V*) and measure the current flowing through the bilayer (*I*), Ag/AgCl electrodes with 1.5% agarose/2 M KCl bridges were used. “Positive voltage” refers here to the case where the *cis* side compartment is positive with respect to the *trans* side. The transmembrane current was measured using an Axopatch 200B amplifier (Molecular Devices, LLC, Orleans Drive, Sunnyvale, CA, USA) in the voltage clamp mode. Data were digitized with a Digidata 1440A and analyzed in pClamp 10.0 (Molecular Devices, LLC, Orleans Drive, Sunnyvale, CA, USA) and Origin 8.0 (OriginLab Corporation, Northampton, MA, USA). Data were acquired at a sampling frequency of 5 kHz using low-pass filtering at 1 kHz, and the current tracks were processed using an 8-pole Bessel at 100 kHz filter.

The number of the PMB molecules involved in the formation of conductive units (*m*) was determined as a slope of the linear regression of the growth part of bilogarithmic plot of the dependence of lipopeptide-induced steady-state transmembrane current on its concentration at 50 mV.

The two-sided addition of phloretin, vardenafil, RH 421, tetracaine, and LPC (from stock ethanol/methanol/DMSO/water mM solutions) up to concentrations of 20, 100, 10, 500, and 7.5 μM in the bathing solution, respectively, was used to modulate the pore-forming activity of PMB in the DOPC/DOPG/Kdo_2_-Lipid A (49.5/49.5/1 mol%) bilayers. Following the method of [70], oleic acid was added to the both sides of the membrane in the bathing solution at up to 30 μM in the form of a complex with methyl-β-cyclodextrin (1:1) to enable the better delivery and insertion of oleic acid into bilayers. The final concentration of the solvent (ethanol/methanol/DMSO) in the chamber did not exceed 0.1%. The used concentrations of the solvents and 30 μM methyl-β-cyclodextrin did not affect the integrity of the lipid bilayers and did not increase their ion conductance.

The changes in the pore-forming activity of PMB produced by the used small molecule was characterized by the ratio between lipopeptide-induced steady-state transmembrane current after (*I_∞_*) and before (*I_∞_^0^*) the addition of the tested compound at 50 mV.

The mean *m*-values and *I_∞_/I_∞_^0^*-ratios were averaged from 3 to 10 bilayers and are presented as mean ± standard deviation (*p* ≤ 0.05).

### 3.3. Electrophysiological Measurements of Changes in Membrane Boundary Potential Induced by Agonists and Antagonists of PMB Pore Formation

The changes in the K^+^-nonactin steady-state bilayer conductance were measured to estimate the changes of the boundary potential of membranes composed of DOPC/DOPG/Kdo_2_-Lipid A (49.5/49.5/1 mol%) (*Δϕ_b_*) after the two-sided addition of phloretin, vardenafil, RH 421, and tetracaine at up to 20, 100, 10, and 500 μM in the bathing solution (0.1 M KCl and 5 mM HEPES at pH 7.4), respectively. Calculations of *Δφ_b_* were performed and suggested that the bilayer conductance was related to the potential drop between the aqueous solution and the bilayer hydrophobic core by the Boltzmann distribution [67]:(1)$$\Delta \phi_{b}=\;\left(kT/e\right)\ln\left(G_{m}/G_{m}^{0}\right)$$
where *G_m_* and *G_m_^0^* are the steady-state membrane conductance induced by K^+^-nonactin in the presence and absence of small molecules, respectively; *ze* is the ion charge; *k* is the Boltzmann constant; and *T* is the absolute temperature.

The mean values of *Δφ_b_* were averaged from 3 to 5 lipid bilayers and are presented as mean ± standard deviation (*p* ≤ 0.05).

### 3.4. Confocal Fluorescence Microscopy

Giant unilamellar vesicles were prepared from a mixture of 1-palmitoyl-2-oleoyl-glycero-3-phosphocholine/1-palmitoyl-2-oleoyl-*sn*-glycero-3-phospho-(1’-rac-glycerol)/Kdo_2_-Lipid A (79/20/1 mol%) and 1 mol% of a fluorescent lipid probe, Rh-DPPE, via the electroformation method (standard protocol of 1 h, 3 V, 10 Hz, and 25 °C) using Nanion Vesicle Prep Pro (Munich, Germany). The resulting liposome suspension contained 0.5 mM of lipids in a 0.5 M sorbitol solution. Up to 5 μM of polymyxin B was added to liposome suspension alone and in combination with 400 μM of phloretin, and vesicles were incubated at room temperature (25 ± 1 °C) for 30 min. At a concentration of 400 μM, phloretin had no independent effect on the lipid vesicles. The vesicles were imaged through an oil immersion objective (60×/1.4HCX PL) using an Apo Leica TCS SP5 confocal laser system (Leica Microsystems, Mannheim, Germany). A helium–neon laser with a wavelength of 561 nm was used to excite Rh-DPPE. The temperature during observation was controlled by air heating/cooling in a thermally insulated camera.

## 4. Conclusions

In summary, we concluded that:(i)The type of negatively charged phospholipid (DOPG vs DOPS) does not affect the pore-forming activity of polymyxin B;(ii)The pore-forming activity of polymyxin B depends on the shape of the membrane lipids;(iii)Polymyxin B is assumed to produce toroidal lipopeptide-lipid pores;(iv)Polymyxin pores in DOPG and Kdo_2_-Lipid A membranes are characterized by different stoichiometries: dimers and hexamers are involved in pore formation in the absence and in the presence of lipopolysaccharides, respectively;(v)Small molecules diminishing membrane dipole potential and inducing positive curvature stress are agonists of pore formation by polymyxin B.

The elucidation of the significance of regulatory factors such as spontaneous curvature stress and membrane dipole potential for the formation and functioning of polymyxin B pores might have decisive consequences for the development of innovative formulations of the studied antibiotic.

## Figures and Tables

**Figure 1 antibiotics-11-01445-f001:**
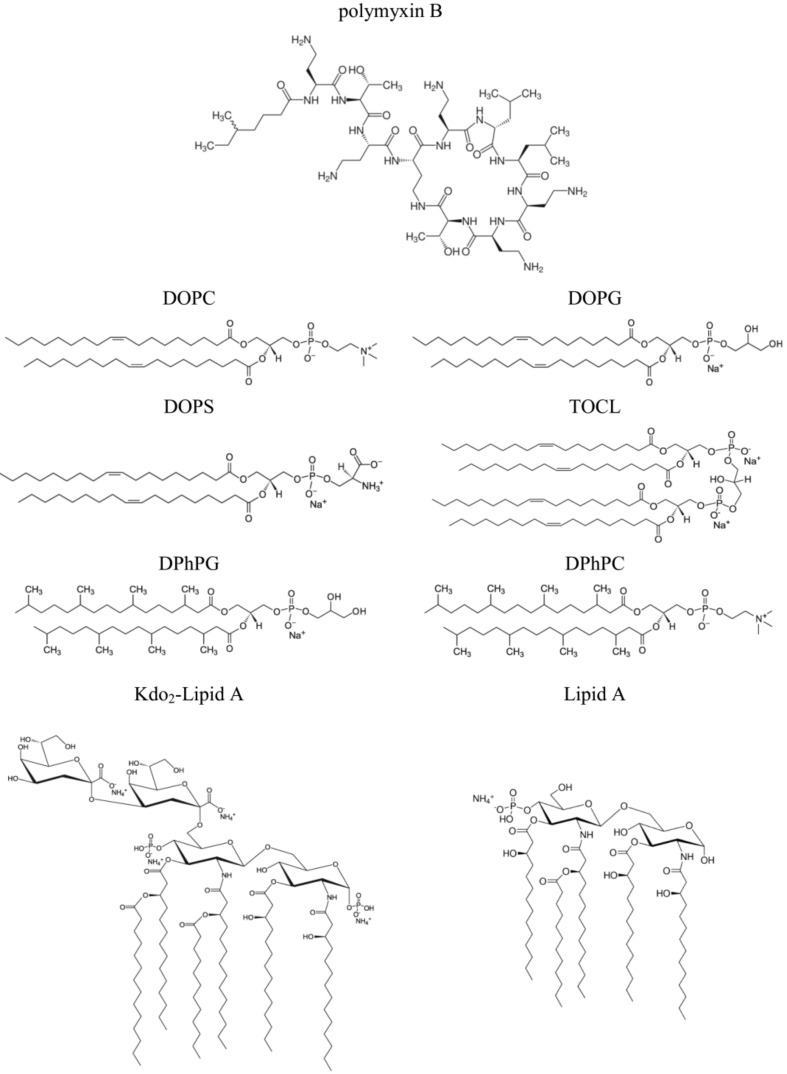
The chemical structures of polymyxin B and lipids used to form model membranes: dioleoylphosphocholine (DOPC), dioleoylphosphoglycerol (DOPG), dioleoylphosphoserine (DOPS), cardiolipin (TOCL), diphytanoylphosphoglycerol (DPhPG), diphytanoylphosphocholine (DPhPC), di [3-deoxy-D-manno-octulosonyl]-lipid A (Kdo_2_-Lipid A), and Lipid A.

**Figure 2 antibiotics-11-01445-f002:**
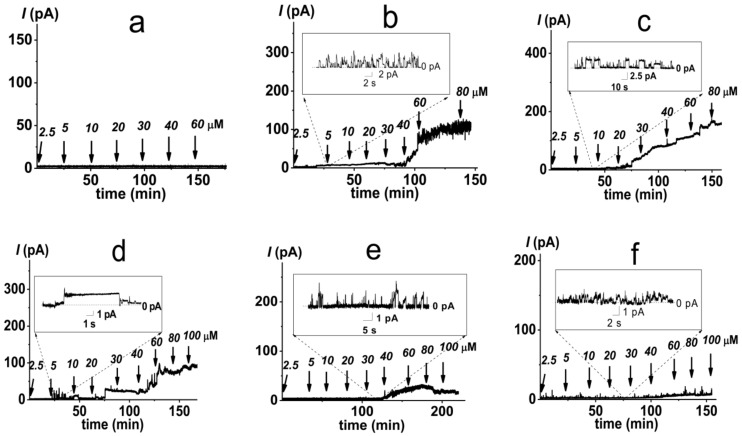
Examples of time tracks of polymyxin B-induced current (*I*) flowing through membranes composed of different phospholipids: DOPC (**a**), DOPG (**b**), DOPC/DOPG (50/50 mol%) (**c**), DOPS (**d**), DPhPG (**e**), and TOCL (**f**). The bilayers were bathed in 0.1 M KCl, pH 7.4. The transmembrane voltage was equal to 50 mV. The moments of polymyxin B addition are indicated by arrows. Above the arrows are the concentrations of the antibiotic in the membrane-bathing solution. Insets: Current fluctuations corresponding to openings and closures of single polymyxin B pores in the membranes of appropriate composition.

**Figure 3 antibiotics-11-01445-f003:**
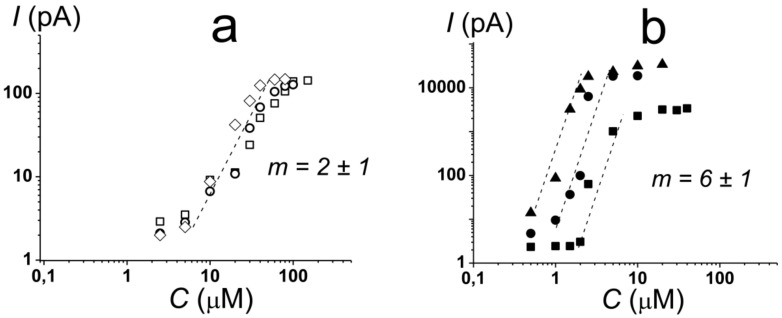
Dependence of the steady-state polymyxin B-induced transmembrane current (*I*) on the concentration of lipopeptide (*C*) in bilogarithmic coordinates. The membranes were made from (**a**) DOPG (**□**), DOPS (◊), and DOPC/DOPG (50/50 mol%) (**○**) and from (**b**) DOPC/DOPG/Kdo_2_-Lipid A (49.5/49.5/1 mol%) (**▲**), DOPC/Kdo_2_-Lipid A (99/1 mol%) (**●**), and DPhPC/DPhPG/Kdo_2_-Lipid A (49.5/49.5/1 mol%) (■), and they were bathed in 0.1 M KCl, pH 7.4. The transmembrane voltage was equal to 50 mV. Slopes of the dependences characterize the number of polymyxin B molecules involved in the pore formation (*m*).

**Figure 4 antibiotics-11-01445-f004:**
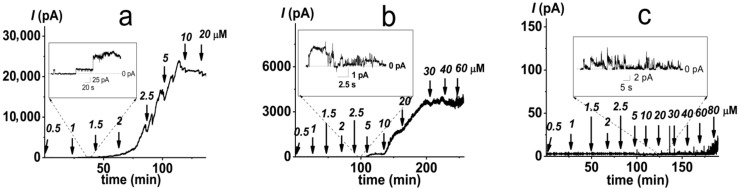
Examples of time tracks of polymyxin B-induced current (*I*) flowing through membranes composed of DOPC/DOPG/Kdo_2_-Lipid A (49.5/49.5/1 mol%) (**a**), DPhPC/DPhPG/Kdo_2_-Lipid A (49.5/49.5/1 mol%) (**b**), and DOPC/DOPG/Lipid A (49.5/49.5/1 mol%) (**c**). The bilayers were bathed in 0.1 M KCl, pH 7.4. The transmembrane voltage was equal to 50 mV. The moments of polymyxin B addition are indicated by arrows. Above the arrows are the concentrations of the antibiotic in the membrane-bathing solution. *Insets:* Current fluctuations corresponding to openings and closures of single polymyxin B channels in the membranes of appropriate composition.

**Figure 5 antibiotics-11-01445-f005:**
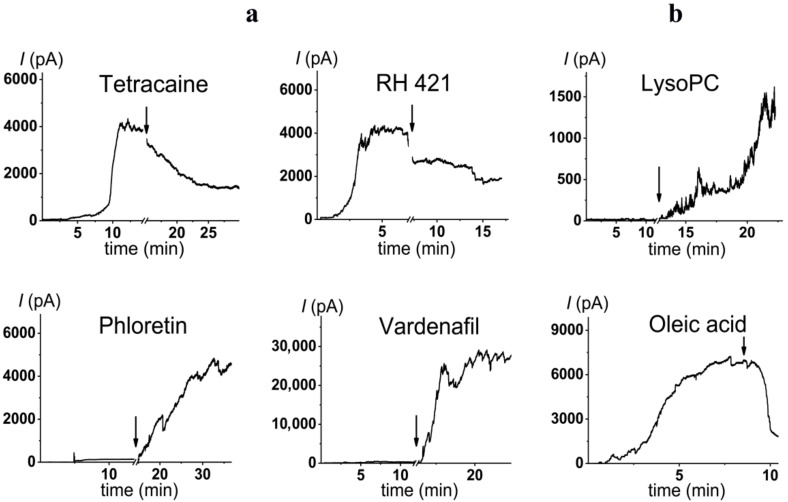
The effects of potential agonists and antagonists on the polymyxin B-induced macroscopic current (*I*). The moments of two-sided addition of small molecules modifying the membrane boundary potential (panel **a**)—tetracaine up to 500 μM, RH 421 up to 10 μM, phloretin up to 20 μM, and vardenafil up to 100 μM—and altering the membrane spontaneous curvature stress (panel **b**)—palmitoylphosphocholine (LysoPC) up to 7.5 μM and oleic acid* up to 30 μM—to the bilayer bathing solution are indicated by the arrows. The membranes were made from DOPC/DOPG/Kdo_2_-Lipid A (49.5/49.5/1 mol%) and bathed in 0.1 M KCl, pH 7.4. The transmembrane voltage was equal to 50 mV. * Oleic acid was added to the bathing solution in the form of a complex with methyl-β-cyclodextrin (1:1) to improve the delivery and insertion of oleic acid into the membrane (Materials and Methods).

**Figure 6 antibiotics-11-01445-f006:**
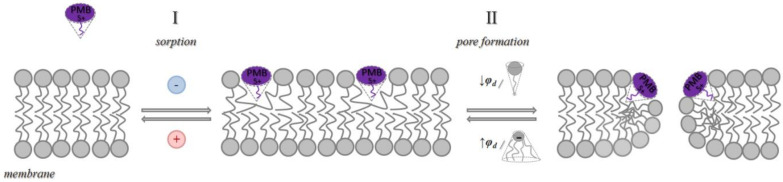
The schematic representation of the pore formation by PMB. At least two distinct steps are proposed. Step I is the adsorption of the lipopeptide onto the membrane, which depends on the charge of the membrane-forming lipids and incorporated molecules; step II is the formation of toroidal lipopeptide/lipid pores, which involves PMB immersion into the bilayer and the bending of opposite lipid monolayer and which depends on the membrane dipole potential and the shape of lipids and incorporated molecules.

**Figure 7 antibiotics-11-01445-f007:**
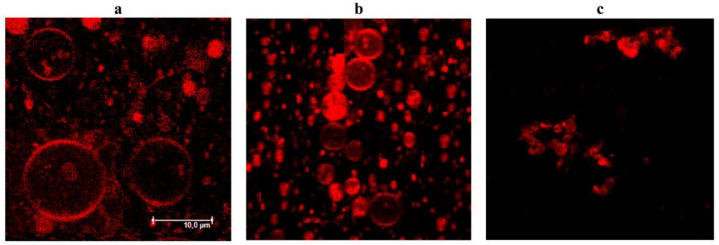
Fluorescence micrographs of giant unilamellar vesicles composed of palmitoyloleoylphosphocholine/palmitoyloleoylphosphoglycerol/Kdo_2_-Lipid A (79/20/1 mol%) in the absence of any modifiers (**a**) and in the presence of 5 µM of polymyxin B alone (**b**) and in combination with 400 µM of phloretin (**c**). The scale bar is shown in panel a.

**Table 1 antibiotics-11-01445-t001:** The action of polymyxin B on bacterial membrane targets.

Potential Target	Proposed Molecular Mechanisms	References
Outer membrane	Binding to lipopolysaccharides, altering lipid packing, permeabilization and/or self-promoted diffusion	[13,14,15,16,17,18,19,20,21,22,23,24,25,26,27,28,29,30,43,44,45]
Contacts between outer and inner membranes	Inducing the formation of contacts and lipid exchange	[27,46,47,48]
Inner membrane	Binding to acidic phospholipids, altering lipid packing, and formation of pores	[34,35,36,37,38]

**Table 2 antibiotics-11-01445-t002:** The effect of small molecules modifying the electrical properties of the lipid bilayers on the polymyxin B-induced transmembrane current.

Small Molecule	Chemical Structure	C (µM)	I_∞_/I_∞_^0^	Δφ_b_ (mV)	Charge ^#^ (%)
phloretin	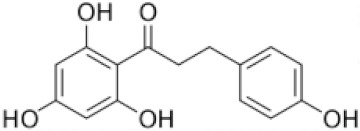	20	28 ± 4	−75 ± 10	23
vardenafil	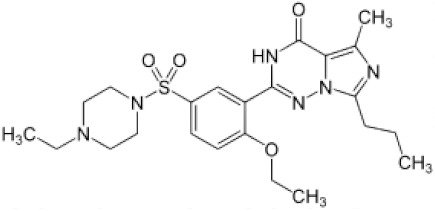	100	49 ± 8	−60 ± 25	6
RH421	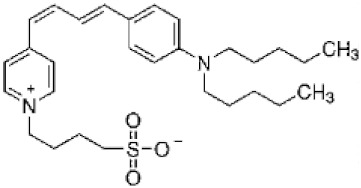	10	0.5 ± 0.1	80 ± 40	3
tetracaine	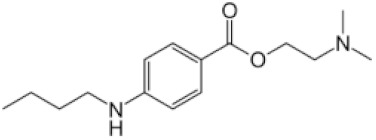	500	0.3 ± 0.2	55 ± 10	91

*I_∞_/I_∞_^0^*—ratio between the PMB-induced transmembrane current in the presence and absence of the small molecule. Membranes were composed of DOPC/DOPG/Kdo_2_-Lipid A (49.5/49.5/1 mol%). *V* = 50 mV. Δφ*_b_*—the changes in the boundary potential of the DOPC/DOPG/Kdo_2_-Lipid A (49.5/49.5/1 mol%) membranes in the presence of tested compounds at indicated concentrations. ^#^ the portion of molecules in the charged form at pH 7.4 was predicted by ChemAxon.

**Table 3 antibiotics-11-01445-t003:** The effect of small molecules inducing membrane curvature stress on polymyxin B-induced transmembrane current.

Modifier	Chemical Structure	C (µM)	I_∞_/I_∞_^0^	R (Å)
LysoPC	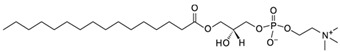	7.5	32 ± 10	68 [77]
Oleic acid	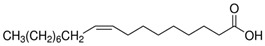	30	0.4 ± 0.2	–25.4 * [78]

*I_∞_/I_∞_^0^*—ratio between the polymyxin B-induced transmembrane current in the presence and absence of the modifier. Membranes were composed of DOPC/DOPG/Kdo_2_-Lipid A (49.5/49.5/1 mol%). *V* = 50 mV. *R—*spontaneous radii of curvature (*—the value for equimolar mixture of DOPC and Oleic acid).

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
