# Peer review of "Lipid Microenvironment Modulates the Pore-Forming Ability of Polymyxin B"

_antibiotics, 2022, doi:10.3390/antibiotics11101445_

Round 1
Reviewer 1 Report
The manuscript analysis the polymyxin action on in vitro prepared membrane composed of different phospholipids and LPS. The results are interesting and suggest the pore formation with this important antibiotic.
Minor suggestions:
Writing of lat. bacterial names: when mentioned for the first, the term has to be spelled out with genus and species, the spp. in not in italic.
Author Response
Reviewer 1 general comment.
The manuscript analyses the polymyxin action on in vitro prepared membrane composed of different phospholipids and LPS. The results are interesting and suggest the pore formation with this important antibiotic.
Minor comment. Writing of lat. bacterial names: when mentioned for the first, the term has to be spelled out with genus and species, the spp. in not in italic.
Answer. We highly appreciate the Reviewer for the careful reading; the unfortunate mistake has been corrected (page 1, lines 28-29).
Reviewer 2 Report
The manuscript by Zakharova et al covers an interesting topic, the research is well conducted and the paper is well written.
However, the reference list must be updated with more recent publications, as many of the cited articles are very old.
Author Response
Reviewer 2 general comment.
The manuscript by Zakharova et al. covers an interesting topic, the research is well conducted and the paper is well written.
Comment. However, the reference list must be updated with more recent publications, as many of the cited articles are very old.
Answer. We highly appreciate the Reviewer for the careful reading; and a new Table 1 summarizing the literature data concerning the mechanisms of antibacterial activity of PMB has been included into Introduction section, and the references list has been updated accordingly (page 3, lines 87-91; pages 12-14, lines 2, 9, 39-48, 58).
Reviewer 3 Report
The submitted manuscript entitled "Lipid microenvironment modulates the pore-forming ability of polymyxin B" is an interesting experimental project involving the well-known antibiotic polymyxin B.
Utilizing its crucial activity of forming ion pores, the interactions of polymyxin B in different environments was investigated.
The manuscript could be considered original in its field and the overall merit is acceptable and suitable for a future publication in Antibiotics.
The manuscript is relevant in the antimicrobial field, even if it could be improved in some areas to make it more clear.
In the Introduction section, when the discussion of polymyxin B starts, it is interesting to have a Figure describing at least the 2D ChemDraw structure of the antibiotic, as it is the crucial structure for the manuscript.
Sticking to the Introduction section, it could be interesting to expand the description of LPS, in its composition and portions, as then Kdo and Lipid A are mentioned many times through the manuscript. An explicative figure of LPS is suggested.
In the Introduction, all the effect of PMB are listed and well explained. A sum-up in a small Table is suggested.
Remember to use a unique style when it comes of terminology of bacteria. For example, let's stick to "Gram-negative" and not sometimes "gram-negative" or other forms.
From the Abstract and throughout the manuscript, it is not clear what you intend as "toroidal pores". It is interesting to write it down the meaning.
As already mentioned, Figures and Tables could be in general increased in number and improved.
In Figure 1, explain the meaning od DOPC (maybe in the caption). Moreover, the spectra reported are too little, and also the resolution could be improved.
In Figure 1, table 1, and 2, better use "C (uM)" than "C, uM". This is the same in Figure 3 "t (min)"and "I (pA)".
In Figure 2, explain in the caption what is reported in x- and y- axes.
For Figure 3, it is interesting to report also the 2D ChemDraw structures of the cited molecules.
In line 24, remember when you write "N-acetylated", "N-" has to be in italics.
In line 28, you mention for the first time "E. coli". As for the other bacteria cited in the manuscript, write it for the first time with a complete name, e.g. Escherichia coli, then stick to the abbreviation.
This is the same suggestion I give for DOPC - DOPG - DPhPC - DPhPG - Kdo - DOPS, in particular in the Results and Discussion section.
In line 141,explain better what you mean for "open circles".
In line 162, explain why you mention "suringomycin E".
References are well reported, up-to-date and filled with all the required information to easily retrieve the desired citation. For the sake of completes, just add the doi in Ref. 38.
In general, be careful to the mix of British and American English through the entire manuscript.
A final section of "Abbreviations" could be very explicative.
For the Conclusion, the schematic way of writing is appreciated, but it is interesting to have final sentences as remarks of the orginality and novelty of the presented work
I suggest looking forward to the publication process, and accepting the manuscript after the required Minor Revisions.
Author Response
Reviewer 3 general comment.
The submitted manuscript entitled "Lipid microenvironment modulates the pore-forming ability of polymyxin B" is an interesting experimental project involving the well-known antibiotic polymyxin B.
Utilizing its crucial activity of forming ion pores, the interactions of polymyxin B in different environments was investigated.
The manuscript could be considered original in its field and the overall merit is acceptable and suitable for a future publication in Antibiotics.
The manuscript is relevant in the antimicrobial field, even if it could be improved in some areas to make it more clear.
Comment 1. In the Introduction section, when the discussion of polymyxin B starts, it is interesting to have a Figure describing at least the 2D ChemDraw structure of the antibiotic, as it is the crucial structure for the manuscript.
Answer 1. According to the Reviewer's suggestion, we have added a Figure with the chemical structure of the antibiotic (new Figure 1).
Comment 2. Sticking to the Introduction section, it could be interesting to expand the description of LPS, in its composition and portions, as then Kdo and Lipid A are mentioned many times through the manuscript. An explicative figure of LPS is suggested.
Answer 2. According to the Reviewer's comment, we have supplemented the new Figure 1 with chemical structures of all lipids used in the study and added the brief description of LPS chemistry in the Introduction (pages 1-2, lines 40-45).
Comment 3. In the Introduction, all the effect of PMB are listed and well explained. A sum-up in a small Table is suggested.
Answer 3. According to the Reviewer's suggestion, we have supplemented the Introduction with new Table 1 summarizing the membrane-associated effects of PMB, the reference list has been also updated (page 3, lines 90-91; pages 13-14, lines 39-48).
Comment 4. Remember to use a unique style when it comes of terminology of bacteria. For example, let's stick to "Gram-negative" and not sometimes "gram-negative" or other forms.
Answer 4. We highly appreciate the Reviewer for the careful reading; the inaccuracies have been corrected throughout the text.
Comment 5. From the Abstract and throughout the manuscript, it is not clear what you intend as "toroidal pores". It is interesting to write it down the meaning.
Answer 5. According to the Reviewer's suggestion, we have supplemented the appropriate paragraph with the explanation of what is a toroidal lipopeptide-lipid pore (page 6, lines 179-190).
Comment 6. As already mentioned, Figures and Tables could be in general increased in number and improved.
Answer 6. According to the Reviewer's comment, we have restructured the Figure 1 to improve the resolution (new Figure 2 and 4), supplemented the Table 1 and 2 (new Tables 2 and 3) with chemical structures of small molecules used, and added new Figure 1 with chemical structures of PMB and lipids, and new Table 1 summarizing the membrane-associated effects of PMB.
Comment 7. In Figure 1, explain the meaning od DOPC (maybe in the caption). Moreover, the spectra reported are too little, and also the resolution could be improved.
Comment 8. In Figure 1, table 1, and 2, better use "C (uM)" than "C, uM". This is the same in Figure 3 "t (min)"and "I (pA)".
Comment 9. In Figure 2, explain in the caption what is reported in x- and y- axes.
Comment 10. For Figure 3, it is interesting to report also the 2D ChemDraw structures of the cited molecules.
Answer to comments (7-10). Abbreviations for lipids and small molecules used have been included in the captions of new Figure 1 and Figure 3 (new Figure 5), and in the text, when mentioned for the first time. We have restructured the Figure 1 to improve the resolution (new Figure 2 and 4), and revised Figure 4 (new Figure 6), Table 1 (new Table 2), and Table 2 (new Table 3). The captions have been also revised to explain what is reported in x- and y- axes.
Comment 11. In line 24, remember when you write "N-acetylated", "N-" has to be in italics.
Comment 12. In line 28, you mention for the first time "E. coli". As for the other bacteria cited in the manuscript, write it for the first time with a complete name, e.g. Escherichia coli, then stick to the abbreviation.
Answer 11-12. We highly appreciate the Reviewer for the careful reading; the inaccuracies have been corrected (page 1, lines 25, 28).
Comment 13. This is the same suggestion I give for DOPC - DOPG - DPhPC - DPhPG - Kdo - DOPS, in particular in the Results and Discussion section.
Answer 13. According to the Reviewer’s suggestion, the abbreviations for lipids used have been included in the caption of new Figure 1 and in the text, when mentioned for the first time. (page 1, line 44; page 2, lines 54-57; page 4, lines 107, 112, 123-124, page 5, line 168, page 6, lines 175-176, 195-196, page 7, lines 237-238, page 9, lines 326-327).
Comment 14. In line 141, explain better what you mean for "open circles".
Answer 14. According to the Reviewer’s comment, we have revised the appropriate sentences (page 5, lines 135, 139, 158; page 6, lines 205, 221; page 7, line 240).
Comment 15. In line 162, explain why you mention "suringomycin E".
Answer 15. According to the Reviewer’s comment, we have revised the paragraph (page 6, lines 179-190).
Comment 16. References are well reported, up-to-date and filled with all the required information to easily retrieve the desired citation. For the sake of completes, just add the doi in Ref. 38.
Answer 16. We highly appreciate the Reviewer for the careful reading; however, the Ref. 38 can not be identified with DOI.
Comment 17. In general, be careful to the mix of British and American English through the entire manuscript.
Answer 17. We highly appreciate the Reviewer for the careful review of the text; some language edits have been made.
Comment 18. A final section of "Abbreviations" could be very explicative.
Answer 18. According to the Reviewer's comment, we have supplemented the manuscript with Abbreviation section (page 12, lines 433-438).
Comment 19. For the Conclusion, the schematic way of writing is appreciated, but it is interesting to have final sentences as remarks of the orginality and novelty of the presented work
Answer 9. According to the Reviewer's suggestion, we have supplemented the Conclusions with remark concerning novelty and significance of the findings (page 12, lines 428-431).
Comment 20. I suggest looking forward to the publication process, and accepting the manuscript after the required Minor Revisions.
Answer 20. We are flattered the Reviewer by the careful reading of the manuscript and. We also thank the Reviewers for comprehensive comments helped us to improve the manuscript.
Reviewer 4 Report
The authors of the manuscript titled "Lipid microenvironment modulates the pore-forming ability of polymyxin B" report the pore-forming ability of polymyxin B and how the lipid environment drastically influences the pore-forming ability. The body of work presented here is appropriate for Antibiotics; however, it needs some major revisions before it can be considered for publication.
Points that need to be addressed.
1. The whole study will provide value if the authors can do a combination studies of polymyxin B with other antibiotics. If the pore-forming ability of PMB can considerably reduce the effective MIC of other antibiotics, it will be a step in the right direction. Additionally, in section 2.3, the authors emphasized the use of compounds that can diminish the dipole potential of the membrane can be used along with PMB. In order to validate this point, the authors need to provide data for combination studies of PMB with these small molecules against bacterial strains along with the transmembrane current measurement.
2. Line 30, there should be an and between the bacterial strains.
3. The authors need to provide a better-quality figure 1 and make sure to enlarge the panels and move the letters a, b, c, d, e, f, g, h, i inside the panels to save space. Also, label the y-axis.
4. Line 114, it should be figure 1b instead of 1a.
5. In the case of measuring the current (figure 1), for each of the bilipid layers, why did the authors not add the PMB at the same interval for each of the experiment? If they can provide experiments with the same concentrations added across the same time interval against all the bilipid layers, it will provide a better conclusion for the readers. Were the experiments done in duplicates or triplicates?
6. Have the authors thought about incubating the bilipid layers with PMB for a while before measuring the current? Do you think the incubation period will provide PMB with the time required for the pore-formation, which will influence the transmembrane current?
7. Have the authors thought about providing microscopic evidence along with the transmembrane current reading, which will substantiate your work?
- In section 3, materials and methods, the authors need to provide information regarding whether the experiments were performed in duplicates or triplicates. The number of independent experiments should be given and provide information regarding whether these numbers are replicates within a single assay.
Author Response
Reviewer 4 general comment.
The authors of the manuscript titled "Lipid microenvironment modulates the pore-forming ability of polymyxin B" report the pore-forming ability of polymyxin B and how the lipid environment drastically influences the pore-forming ability. The body of work presented here is appropriate for Antibiotics; however, it needs some major revisions before it can be considered for publication.
Comment 1. The whole study will provide value if the authors can do a combination studies of polymyxin B with other antibiotics. If the pore-forming ability of PMB can considerably reduce the effective MIC of other antibiotics, it will be a step in the right direction. Additionally, in section 2.3, the authors emphasized the use of compounds that can diminish the dipole potential of the membrane can be used along with PMB. In order to validate this point, the authors need to provide data for combination studies of PMB with these small molecules against bacterial strains along with the transmembrane current measurement.
Answer 1. We are grateful to the Reviewer for raising this important issue, the significance of which we understand very well. Currently, the study of the synergism of the antimicrobial action of polymyxin B and the small molecules that potentiate its pore-forming ability is being actively carried out. Much needs to be done to test the synergy on a broad panel of human pathogens. Moreover, we are also interested to answer the question by the molecular mechanisms of the synergism of the antibacterial action of polymyxin B with other small molecules. For example, Liu et al. (BMC Microbiology, 2020, 20, 306, doi: 10.1186/s12866-020-01995-1) have found that resveratrol (plant polyphenol having structure similar to phloretin) enhances the antimicrobial effect of polymyxin B on Klebsiella pneumoniae and Escherichia coli. Taking into account that resveratrol dramatically affects the thermotropic behavior of membrane lipids and is assumed to modulate the spontaneous curvature stress in the bilayer, the elucidation of the role of these effects in the potentiating the polymyxin B activity by resveratrol should be of key importance. We hope that the data concerning the synergism of antibacterial action of polymyxin B with various small molecule agonists will form the basis for our next publication.
Comment 2. Line 30, there should be an and between the bacterial strains.
Answer 2. The inaccuracy noted by the Reviewer has been eliminated.
Comment 3. The authors need to provide a better-quality figure 1 and make sure to enlarge the panels and move the letters a, b, c, d, e, f, g, h, i inside the panels to save space. Also, label the y-axis.
Answer 3. According to the Reviewer's suggestion, we have restructured the Figure 1 to improve the resolution (new Figure 2 and 4). The captions have been also revised to explain what is reported in x- and y- axes.
Comment 4. Line 114, it should be figure 1b instead of 1a.
Answer 4. We highly appreciate the Reviewer for the careful reading; the unfortunate mistake has been corrected.
Comment 5. In the case of measuring the current (figure 1), for each of the bilipid layers, why did the authors not add the PMB at the same interval for each of the experiment? If they can provide experiments with the same concentrations added across the same time interval against all the bilipid layers, it will provide a better conclusion for the readers. Were the experiments done in duplicates or triplicates?
Comment 6. Have the authors thought about incubating the bilipid layers with PMB for a while before measuring the current? Do you think the incubation period will provide PMB with the time required for the pore-formation, which will influence the transmembrane current?
Answer to comments 5-6. The experiments were done in triplicates to tenplicates (3÷10 independent experiments). The observed effects (quantifiable as the transmembrane current at appropriate polymyxin concentration in membranes of different compositions and the ratio between currents before and after small molecule addition) did not depend on the moment at which the antibiotic was added (the incubation time) (if the current steady-state level had been already reached, it usually took 10-40 min, the criterion for reaching a plateau was dI/dt≤0 for several minutes). But in order not to confuse the readers, we have replaced some figures with tracks representing experiments with the introduction of the antibiotic across the similar time intervals (please, see new Figure 2 for phospholipid membranes and Figure 4 for bilayers containing lypopolysaccharides).
Comment 7. Have the authors thought about providing microscopic evidence along with the transmembrane current reading, which will substantiate your work?
Answer 7. According to the Reviewer's and taking into account the data concerning the influence of the antibiotic on the contacts between the outer and inner membranes of Gram-negative bacteria (Clausell et al., J Phys Chem B, 2007, 111(3), 551-563, doi: 10.1021/jp064757+; Cajal et al., Biochemi, 1996, 35(1), 299–308, doi: 10.1021/bi9512408; Clausell et al., Luminescence, 2005, 20(3), 117–123, doi: 10.1002/bio.810; Clausell et al., J Phys Chem B, 2006, 110(9), 4465–4471, doi: 10.1021/jp0551972), we have made an attempt to perform preliminary experiments using confocal fluorescence microscopy.
Giant unilamellar vesicles were prepared from mixture of POPC/POPG/Kdo2-Lipid A (79/20/1 mol %) and 1 mol% of fluorescent lipid probe, dipalmitoylphosphoethanolamine-N-lissamine rhodamine, by the electroformation method (standard 1 hour-protocol of formation at 3 V, 10 Hz, and 25 °C) using Nanion Vesicle Prep Pro (Munich, Germany). The resulting liposome suspension contained 0.5 mM lipid in 0.5 M sorbitol solution. The measurements of the liposome were added polymyxin B up to different concentrations (1, 5 and 20 μM) alone and with phloretin (40 and 400 μM) at room temperature (25 ± 1 °C). Vesicles were imaged through an oil immersion objective (65×/1.4HCX PL) using an Olympus (Germany). A helium-neon laser with a wavelength of 561 nm was used to excite Rh-DPPE. The temperature during observation was controlled by air heating/cooling in a thermally insulated camera.
Figure R1 (please see attached file) presents the fluorescence micrographs of POPC/POPG/Kdo2-Lipid A liposomes in the absence (control) and presence of polymyxin B alone at the concentrations 1 and 5 µM. Increase in the antibiotic concentration led to aggregation of liposomes and decrease in their size. An addition of phloretin potentiated the effects of polymyxin B. Combination of polymyxin B at 5 μM and phloretin at 400 μM caused the complete destruction and aggregation of POPC/POPG/Kdo2-Lipid A vesicles, demonstrating that phloretin enhanced the membrane lytic properties of polymyxin B.
Comment 8. In section 3, materials and methods, the authors need to provide information regarding whether the experiments were performed in duplicates or triplicates. The number of independent experiments should be given and provide information regarding whether these numbers are replicates within a single assay.
Answer 8. According to the Reviewer's suggestion, we have revised this part of Materials and Methods section (page 11, lines 400-401). The mean m-values and I∞/I∞0-ratios were averaged from 3 to 10 bilayers and presented as mean ± standard deviation (p ≤ 0.05 The mean values of Δφb were averaged from 3 to 5 lipid bilayers and presented as mean ± standard deviation (p ≤ 0.05).

Round 2
Reviewer 4 Report
The authors have addressed most of the comments. The corresponding text, figure, and method description for the fluorescence micrographs should be added to the main text.
Author Response
Reviewer 1 general comment.
The authors have addressed most of the comments. The corresponding text, figure, and method description for the fluorescence micrographs should be added to the main text.
Answer. According to Reviewer's suggestion; the confocal fluorescence data have been included into the manuscript (page 9,10,11,12, lines 320-333, 364-366, 372-373, 433-446).